# Synthesis and Characterization of Vanillin-Based π-Conjugated Polyazomethines and Their Oligomer Model Compounds

**DOI:** 10.3390/molecules27134138

**Published:** 2022-06-28

**Authors:** Lauriane Giraud, Stéphane Grelier, Etienne Grau, Laurent Garel, Georges Hadziioannou, Brice Kauffmann, Éric Cloutet, Henri Cramail, Cyril Brochon

**Affiliations:** 1Laboratoire de Chimie des Polymères Organiques (LCPO), University of Bordeaux, CNRS, Bordeaux INP, 33600 Pessac, France; giraud.lauriane.louise@gmail.com (L.G.); stephane.grelier@u-bordeaux.fr (S.G.); etienne.grau@enscbp.fr (E.G.); hadzii@enscbp.fr (G.H.); cloutet@enscbp.fr (É.C.); 2Solvay—GBU Aroma Performance, Research and Innovation—Centre de Lyon (RICL), 69192 Saint Fons, France; laurent.garel@solvay.com; 3Institut Européen de Chimie Biologie (IECB), University of Bordeaux, CNRS, INSERM, 33600 Pessac, France; b.kauffmann@iecb.u-bordeaux.fr

**Keywords:** polyazomethines, divanillin, bio-based π-conjugated polymers

## Abstract

The synthesis of π-conjugated polymers via an environmentally friendly procedure is generally challenging. Herein, we describe the synthesis of divanillin-based polyazomethines, which are derived from a potentially bio-based monomer. The polymerization is performed in 5 min under microwave irradiation without any metallic catalyst, with water as the only by-product. The vanillin-based polyazomethines were characterized by SEC, TGA, and UV-Vis spectroscopy. Model compounds were designed and characterized by X-ray diffraction and UV-Vis spectroscopy. The structure/properties study of vanillin-based azomethines used as models allowed us to unequivocally confirm the E configuration and to highlight the cross-conjugated nature of divanillin-based polymers.

## 1. Introduction

Semi-conducting π-conjugated polymers are key organic materials for the development of low-cost optoelectronic devices (Organic Light Emitting Diodes, Organic Field Effect Transistor, Organic Photovoltaics) and more particularly for renewable energy production via printing process technologies. These technologies are close to maturity, but they are still impeded by several weaknesses. Indeed, π-conjugated polymers are generally obtained by synthetic methods based on cross-coupling reactions (Stille, Suzuki, Kumada) in harsh conditions that require the use of transition metals (e.g., palladium or nickel) as catalysts, toxic organic solvents and also, in some cases, hazardous monomers which generate stoichiometric amounts of toxic by-products, such as organotin derivatives, that must be removed [1,2]. Consequently, synthetic routes usually rely on complex protocols by resorting to costly and hardly removable catalysts. While efforts are continuously realized to optimize these reactions and to reduce their environmental impacts (use of solvents with a reduced toxicity, etc.) [3], traces of metallic impurities are still found in the final material. Even at a very low content (<20 ppm), it was demonstrated that these impurities have a dramatic effect on the ensuing device’s performance; indeed, they can act as charge traps [4]. Tedious purification processes thus have to be carried out [5,6].

In order to allow these semiconducting organic materials to reach industrial and economic viability, we have been investigating sustainable routes to π-conjugated polymers, which include the use of renewable monomers from biomass and greener processes and syntheses [7,8]. As an example, furane-based π-conjugated polymers have been used successfully for organic solar cells (OSCs) [9]. In particular, we have been developing a new polymer family based on divanillin platforms [10,11,12]. Recently, Arndtsen and collaborators reported an elegant approach to obtain divanillin-based polypyrroles via a multicomponent coupling approach [13]. The latter polymers exhibit strong fluorescence, but the “cross-conjugated” nature of these polymers was not investigated. More recently, in an alternative approach, Leclerc et al. obtained a variety of aromatic brominated monomers which are building blocks for organic electronic materials [14].

In this context, polyazomethines are an attractive class of materials, as the latter can be synthesized by classical polycondensation between diamines and bisaldehydes with the release of water as a by-product. This straightforward synthetic method enables the formation of polymer chains without extensive purification. Conjugated polyazomethines find applications in various fields, such as photovoltaics [15,16,17] and liquid crystals [18]. Until the 1980s, studies reported polyazomethines as poorly soluble in common organic solvents and mainly used as high-stretch fibers [19,20]. The insertion of an alkyl side moiety on the conjugated polyazomethine backbone was a mean to significantly improve solubility in chloroform, toluene, benzene, etc., allowing better structural and physico-chemical characterizations of this class of polymeric materials [21]. Fluorescence, an essential property for various applications in organic electronics, is a major challenge for polyazomethines as the azomethine bond is reported to quench most fluorophores [22]. This issue was overcome in 2008 when Kim et al. [23] first showed that protonation of the azomethine bond could affect the fluorescence properties of conjugated polyazomethines. Since then, it was demonstrated that lowering the temperature of the sample during measurement [24] or a careful choice of the chemical structures (design of copolymers) can also improve the fluorescence properties. In this context a large palette of polyazomethines with various chemical structures have been reported [17,25,26,27].

In the course of designing sustainable conjugated polymers for organic electronic, we report, herein, the synthesis, characterization and properties of a new family of polyazomethines based on divanillin.

## 2. Experimental Section

### 2.1. Materials

Vanillin (>97%), laccase from *Trametes Versicolor*, 2-ethylhexyl bromide (95%), potassium carbonate, para and meta phenylene diamine were obtained from Sigma-Aldrich. Para-toluene sulfonic acid (PTSA, 99%) and 2,7-diamino-9,9-di-*n*-octylfluorene were purchased from TCI. Silica gel (pore size 60 Å, 230–400 mesh particle size, 40–63 µm particle size) was obtained from Honeywell Fluka. Potassium hydroxide and iodomethane were obtained from Fischer. Aniline was obtained from Acros Organics. All products and solvents (reagent grade) were used as received except otherwise mentioned. The solvents were of reagent grade quality and were purified whenever necessary according to the methods reported in the literature.

### 2.2. Instrumentation

^1^H and ^13^C NMR measurements were performed with a Bruker Avance 400 spectrometer (400.20 MHz and 100.7 MHz for ^1^H and ^13^C, respectively) at room temperature using deuterated solvent. Optical absorption spectra were obtained with a UV-visible spectrophotometer (UV-3600, Shimadzu France, Marne-La-Vallée, France). Fluorescence spectra were obtained from a spectrofluorometer (Fluoromax-4, Horiba Scientific, Palaiseau, France). To improve the quality of the spectra, these latter were recorded with higher concentrations to obtain a more intense signal (as the quantum yield is very low) and the excitation was carried out at a lower wavelength than the absorbance maximum. This enables slight shifting of the Raman peak of the solvent, to better see the emission of the polyazomethines. For both absorbance and fluorescence spectra, solvents of spectroscopic grade were used (from Sigma-Aldrich France, Saint Quentin Fallavier, France) and quartz cells were used. Polymer molar masses were determined by size exclusion chromatography (SEC) using a three-columns set of Resipore Agilent: one guard column Resipore Agilent PL1113-1300, then two columns Resipore Agilent PL1113-6300, connected in series, and calibrated with 11 narrow polystyrene standards from polymer Laboratories (375; 945; 1300; 3090; 6660; 12,980; 27,060; 46,380; 107,100; 187,700; 364,000 g/mol, respectively) using both refractometric (GPS 2155) and UV detectors (Viscotek - Malvern Panalytical, Palaiseau, France). THF was used as eluent (0.8 mL/min) and trichlorobenzene as a flow marker (0.15%) at 30 °C. Diffraction data from a single crystal of the different model compounds were measured on a 3 kW microfocus Rigaku FRX rotating anode. The source is equipped with high-flux Osmic Varimax HF mirrors and a hybrid Dectris Pilatus 200 K detector. The source is operating at the copper kα wavelength with a partial χ goniometer that decreases blind areas and enables automatic axial adjustment. Data were processed with the CrysAlisPro suite version 1.171.38.43 [28]. Empirical absorption correction using spherical harmonics, implemented in SCALE3 ABSPACK scaling algorithm was used. The structure was solved with Shelxt [29] and refined by the full-matrix least-squares method on F2 with Shelxl-2014 [29] within Olex2 [30]. For all atoms, anisotropic atomic parameters were used. Hydrogen atoms were place at idealized positions and refined as riding of their carriers with Uiso(H) = 1.2 Ueq (CH, CH_2_, NH) and Uiso(H) = 1.5 Ueq (CH_3_). DFIX and AFIX instructions were used to improve the geometry of molecules and RIGU to model atomic displacement parameters. Disordered solvent molecules were removed using the SQUEEZE procedure from the PLATON suite [31]. For search and analysis of solvent accessible voids in the structures default parameters were utilized: grid 0.20 Å, probe radius 1.2 Å and NStep 6. Calculated total potential solvent accessible void volumes and electron counts per unit cell are given in the CIF files that were checked using IUCR’s checkcif algorithm. Due to the characteristics of some of the crystals, i.e., large volume fractions of disordered solvent molecules, weak diffraction intensity and moderate resolution, few A-level and B-level alerts remain in the check .cif file. These alerts are inherent to the data and refinement procedures and do not reflect errors on the model refined.

### 2.3. Synthesis of Divanillin via Enzymatic Coupling

For the buffer, 5.6 g of sodium acetate and 1.75 mL of acetic acid were dissolved in 1 L of water. In an appropriate vessel, 6 g of vanillin were dissolved in 80 mL of acetone, afterwards 720 mL of the previously prepared buffer were added. Furthermore, 49.6 mg of Laccase from *Trametes Versicolor* were finally added and the medium was saturated with oxygen. After 24 h at 25 °C under constant and light stirring, the medium was filtrated to give crude divanillin as a brown powder—the filtrate was charged again in vanillin and saturated in oxygen to start a new cycle. To purify it further, 6 g of crude divanillin was completely dissolved in 100 mL of a 0.5 M solution of NaOH. This solution was then poured in 600 mL of ethanol, which was then acidified with fuming HCl until divanillin spontaneously precipitated. It was then recovered by simple filtration and rinsed with water and acetone. Yield: 85%. ^1^H NMR (400.20 MHz, DMSO-*d_6_*, δ) 9.81 (s, 2H), 7.44–7.42 (m, 4H), 3.93 (s, 6H). ^13^C NMR (100.70 MHz, DMSO-*d_6_*, δ) 191.2, 150.6, 148.3, 128.2, 127.8, 124.6, 109.2, 56.1. See Appendix A for ^1^H and ^13^C NMR spectra respectively.

### 2.4. Synthesis of Methylated Divanillin (DVM)

In a flame-dried and nitrogen-flushed 100 mL glassware equipped with a condenser, divanillin (6.6 mmol) was solubilized in 40 mL of previously dried DMF. Then K_2_CO_3_ (3.8 g, 27.5 mmol) was added to the reaction mixture, which was heated at 80 °C for 2 h. Then, 2.45 mL of iodomethane (39.5 mmol) were added slowly to the reaction mixture and it was left 16 more hours at 80 °C. Afterwards, the reaction mixture was poured into 300 mL of water and left 5 min until it completely precipitated. The final product was then recovered after simple filtration and extensively dried before use without further purification. Yield: 90%. (This protocol was also used to obtain methylated vanillin.) ^1^H NMR (400.20 MHz, CDCl_3_, δ) 9.84 (s, 2H), 7.44–7.43 (d, *J* = 4 Hz, 2H), 7.33–7.32 (d, *J* = 4 Hz, 2H), 3.91 (s, 6H), 3.70 (s, 6H). ^13^C NMR (100.70 MHz, CDCl_3_, δ) 191.1, 153.5, 152.5, 132.3, 131.9, 127.7, 110.6, 61.1, 56.2. See Appendix A for ^1^H and ^13^C NMR spectra respectively.

### 2.5. Synthesis of 2-Ethylhexylated Divanillin (DVEH)

In a flame-dried and nitrogen flushed 100 mL glassware equipped with a condenser, divanillin (6.6 mmol) was solubilized in 20 mL of previously dried DMSO. Then, KOH (0.89 g, 15.9 mmol) was added to the reaction mixture, which was heated at 80 °C for 2 h. Then, 2.2 equiv of 2-ethylhexyl bromide were added to the reaction mixture and it was left for 16 more hours at 80 °C. Afterwards, the reaction mixture was poured into 300 mL of water and extracted with 100 mL of ethyl acetate three times. The organic phase was dried on MgSO_4_ and the solvent removed using a rotary evaporator. This dried crude was then purified by flash chromatography on a silica column (80 g) using cyclohexane and ethyl acetate as eluents (85/15). Yield: 40%. ^1^H NMR (400.20 MHz, CDCl_3_, δ) 9.82 (s, 2H), 7.4–7.38 (d, *J* = 8 Hz, 4H), 3.88 (s, 6H), 3.78–3.69 (m, 4H), 1.33–0.89 (m, 19H), 0.73 (t, *J* = 8 Hz, 6H), 0.62 (t, *J* = 8 Hz, 6H). ^13^C NMR (100.70 MHz, CDCl_3_, δ) 191.1, 153.6, 152.1, 132.0, 131.6, 128.6, 109.7, 75.2, 55.9, 40.4, 30.3, 29.0, 23.5, 22.9, 14.1, 11.0. See Appendix A for ^1^H and ^13^C NMR spectra, respectively.

### 2.6. Polycondensation of Divanillin Derivatives

In a 10 mL microwave dedicated glassware, a stoichiometric amount of *p*-phenylenediamine, *m*-phenylenediamine or 2,7 diamino(9,9-dioctyl *n*,*n*-fluorene) and of divanillin derivative (either DVM or DVEH) was put in suspension in 5 mL of toluene. Silica (300 mg) was then added with catalytic amount of PTSA and the reaction heated at 130 °C for 4 h using microwave irradiation. Recovery step: the crude polymer was then solubilized in a minimum amount of methylene chloride, and 100 mL of methanol was added. The solution turns cloudy. The solvents are then evaporated using rotary evaporator, to obtain a powder which is rinsed with methanol, to give the final polymer. After optimization: the reaction time was reduced to 5 min without silica, still followed by a recovery step using a rotary evaporator. See Appendix A for ^1^H NMR spectrum, SEC trace and TGA curve for the six polyazomethines obtained. After polymer recovery, the overall yield is about 80%.

### 2.7. Synthesis of Di(3,4-dimethoxyphenylmethylidene)-(1,4-diamine benzene) (M1a)

In a flame-dried flask equipped with stirrer and condenser, 2 equiv of methylated vanillin were dissolved with p-phenylene diamine (1 or 2 equiv) in methanol, under inert atmosphere. The resulting solution was heated at the reflux using conventional heating for 30 min, and then filtrated on 0.45 µm PTFE filter, and the solvent evaporated under vacuum. The obtained product was recrystallized from a warm mixture of methylene chloride and methanol. *E*/*Z* in the crude material: 92/8. Conversion (calculated with ^1^H NMR): 95%, yield after recrystallization: 70%. ^1^H NMR (400 MHz, CD_2_Cl_2_, δ) 8.43 (s, 2H), 7.61 (m, 2H), 7.36–7.33 (d, *J* = 12 Hz, 2H), 7.26 (s, 4H), 6.95 (m, 2H), 3.94 (s, 6H), 3.91 (s, 6H). ^13^C NMR (400 MHz, CD_2_Cl_2_, δ) 159.3, 152.6, 150.2, 150.0, 130.1, 124.5, 122.1, 111.1, 109.5, 56.2. See Appendix A for ^1^H and ^13^C NMR spectra respectively.

### 2.8. Di-(3-methoxy, 4-hydroxyphenylmethylidene)-1,4-diaminobenzene (M1b)

In a flame-dried flask equipped with stirrer and condenser, 2 equiv of vanillin were dissolved with *p*-phenylene diamine (1 equiv) in methanol, under an inert atmosphere. The resulting solution was heated at the reflux using conventional heating for 30 min, and then filtrated on 0.45 µm PTFE filter, and the solvent evaporated under vacuum. The final product was obtained after recrystallization from warm methylene chloride/methanol. *E/Z* in the crude material: 85/15. Conversion (calculated with ^1^H NMR): 85%, yield after recrystallization: 55%. ^1^H NMR (400 MHz, DMSO-*d_6_*, δ) 9.74 (s, 2H), 8.51 (s, 2H), 7.55–7.54 (d, *J* = 4 Hz, 2H), 7.36–7.34 (d, *J* = 8 Hz, 2H), 7.28 (s, 4H), 6.92–6.90 (d, *J* = 8 Hz, 2H), 3.86 (s, 6H). ^13^C NMR (400 MHz, DMSO-*d_6_*, δ) 159.4, 150.2, 148.0, 128.1, 124.1, 121.8, 115.4, 110.4, 55.6. See Appendix A for ^1^H and ^13^C NMR spectra respectively.

### 2.9. Di(3,4-dimethoxyphenylmethylidene)-1,3 diaminobenzene (M2)

In a flame-dried flask equipped with stirrer and condenser, 2 equiv of methylated vanillin were dissolved with *m*-phenylene diamine (1 equiv) in methanol, under inert atmosphere. The resulting solution was heated at the reflux using conventional heating for 30 min, and then filtrated on 0.45 µm PTFE filter, and the solvent evaporated under vacuum. The obtained product was dissolved again in methanol and filtrated on 0.45 µm PTFE. The crystallization starts spontaneously to give the final product. *E/Z* in the crude material: 75/25. Conversion (calculated with ^1^H NMR): 80%, yield after recrystallization: 55%. ^1^H NMR (400 MHz, CD_2_Cl_2_, δ) 8.43 (s, 2H), 7.61–7.60 (d, *J* = 3 Hz, 2H), 7.42–7.34 (m, 3H), 7.07–7.01 (m, 3H), 6.97–6.95 (m, 2H), 3.94 (s, 6H), 3.91 (s, 6H). ^13^C NMR (400 MHz, CD_2_Cl_2_, δ) 160.4, 153.7, 152.7, 150.0, 129.9, 124.6, 118.6, 113.2, 111.1, 109.5, 56.3. See Appendix A for ^1^H and ^13^C NMR spectra respectively.

### 2.10. 1,2-Dimethoxy,4-(phenylimino)methylbenzene (M3a)

In a flame-dried flask equipped with stirrer and condenser, 1 equiv of methylated vanillin was dissolved with aniline (1 equiv) in methanol, under inert atmosphere. The resulting solution was heated at the reflux using conventional heating for 30 min, and then filtrated on 0.45 µm PTFE filter, and the solvent evaporated under vacuum. The obtained product was recrystallized from a warm mixture of methylene chloride and methanol. *E/Z* in the crude material: 100/0. Conversion (calculated with ^1^H NMR): 94%, yield after recrystallization: 70%. ^1^H NMR (400 MHz, CD_2_Cl_2_, δ) 8.37 (s, 1H), 7.59 (d, *J* = 2 Hz, 1H), 7.41–7.37 (m, 2H), 7.35–7.32 (d, *J* = 12 Hz, 1H), 7.24–7.18 (m, 3H), 6.96–6.94 (d, *J* = 8 Hz, 1H), 3.93 (s, 3H), 3.90 (s, 3H). ^13^C NMR (400 MHz, CD_2_Cl_2_, δ) 159.7, 152.3, 129.5, 129.1, 125.5, 124.1, 120.8, 110.7, 109.2, 55.8. See Appendix A for ^1^H and ^13^C NMR spectra respectively.

### 2.11. 2-Methoxy,4-(phenylimino)methylphenol (M3b)

In a flame-dried flask equipped with stirrer and condenser, 1 equiv of vanillin was dissolved with aniline (1 equiv) in methanol, under inert atmosphere. The resulting solution was heated at the reflux using conventional heating for 30 min, and then filtrated on 0.45 µm PTFE filter, and the solvent evaporated under vacuum. The final product was obtained after recrystallization from warm cyclohexane/methylene chloride. E/Z in the crude material: 100/0. Conversion (calculated with ^1^H NMR): 95%, yield after recrystallization: 60%. ^1^H NMR (400 MHz, DMSO-*d_6_*, δ) 9.74 (s, 1H), 8.44 (s, 1H), 7.54–7.53 (d, *J* = 4 Hz, 1H), 7.41–7.36 (m, 2H), 7.35–7.33 (d, *J* = 8 Hz, 1H), 7.21–7.19 (m, 3H), 6.91–6.89 (d, *J* = 8 Hz, 1H), 3.85 (s, 3H). ^13^C NMR (400 MHz, DMSO-*d_6_*, δ) 160.2, 152.1, 150.3, 148.2, 129.2, 128.0, 125.5, 124.4, 120.9, 115.3, 110.4, 55.6. See Appendix A for ^1^H and ^13^C NMR spectra respectively.

### 2.12. Synthesis of 6,6′-Methoxy-5,5′-dimethoxy-[1,1′-biphenyl]-3,3′-di(phenylimino)methyl (M4)

In a flame-dried flask equipped with stirrer and condenser, 1 equiv of DVM was dissolved with aniline (1 equiv) in methanol, under inert atmosphere. The resulting solution was heated at the reflux using conventional heating for 30 min, and then filtrated on 0.45 µm PTFE filter, and the solvent evaporated under vacuum. The obtained product was recrystallized from a warm mixture of methylene chloride and methanol. E/Z in the crude material: cannot be determined clearly. Conversion (calculated with ^1^H NMR): 98%, yield after recrystallization: 75%. ^1^H NMR (400 MHz, CD_2_Cl_2_, δ): 8.42 (s, 2H), 7.67 (d, J = 2 Hz, 2H); 7.42–7.38 (m, 4H), 7.32 (d, J = 2 Hz, 2H), 4.01 (s, 6H), 3.74 (s, 6H). ^13^C NMR (400 MHz, CD_2_Cl_2_, δ): 159.9, 153.7, 152.5, 150.3, 132.7, 132.3, 129.6, 126.2, 125.8, 121.2, 110.5, 61.0, 56.3. See Appendix A for ^1^H and ^13^C NMR spectra, respectively.

## 3. Results and Discussion

### 3.1. Synthesis of Polyazomethines

Alkylated divanillin, either bearing a methyl or a 2-ethylhexyl group, noted DVM or DVEH respectively, was used as the main building block of the so-formed polyazomethines. The latter monomers were reacted at the 1:1 stoichiometry ratio with the three diamines, fluorene diamine, *p*-phenylene diamine (**p-PD**) and *m*-phenylenediamine (**m-PD**) (see Figure 1). The reaction was performed under microwave irradiation, this process enabling shorter reaction times than conventional heating [32,33]. Toluene was used as a solvent as it is one in which the final polyazomethines are fully soluble. PTSA was used as a catalyst and silica as a desiccant, to remove water from the medium and displace the equilibrium towards the formation of the polyazomethines. These experimental conditions do not require any metallic catalysts and the only by-product is water, trapped by silica.

We previously showed that the key step in the process, allowing reduction of the reaction time to only 5 min, is the removal of solvent from the reaction medium [8]. Indeed, depending on the concentration used, silica degrades polyazomethines and hinders their fluorescence (see Appendix A). These improved experimental conditions were successfully applied to the polyazomethines previously described in Figure 1, as exemplified in Figure 1.

The polymer recovery was done by dissolving the crude in a minimum amount of a solvent mixture, which is then evaporated using a rotary evaporator after adding methanol. At this stage, the solution turned turbid but there is no evident precipitation due to the presence of oligomers, which are still soluble enough. Interestingly, the SEC profile of the polyazomethine after recovery shows a single peak towards higher molar masses. It is not due to the removal of short oligomers, but it indicates that the polymerization continues during the recovery step, leading to a shift of the peak in SEC and an improve of conversion.

Polyazomethines can also be formed directly during this step, by applying the recovery procedure to a stoichiometric amount of monomers (see Appendix A).

The formed polyazomethines were characterized and presented in Table 1. They were all soluble in common solvents (chloroform, THF, methylene chloride, toluene), except **P5**, suggesting that use of meta-phenylene diamine leads to steric hindrance along the chain, resulting in polyazomethines exhibiting highly twisted chain conformations. This lack of solubility was not observed for **P6**, possibly because of its lower molar mass and higher dispersity (see Table 1). The polymerization reaction was assessed by ^1^H NMR analysis with the appearance of the characteristic peak of azomethines around 8.4 ppm (see Appendix A).

These divanillin-based polyazomethines exhibit molar masses up to 21,000 g/mol (**P1**, Table 1). They all exhibit a dispersity close to 2 in agreement with a polycondensation methodology. It is worth noticing that polyazomethines from DVEH have higher molar masses than the ones from DVEM (See **P1**/**P2** and **P3**/**P4** in Table 1). Such a feature can be explained by a difference of solubility. Indeed, polyazomethines from DVEH remain longer in solution in comparison to the ones from DVEM, thus leading to higher molar mass polyazomethines in comparison to their DVEM-based homologues. The molar masses could also be estimated from ^1^H NMR analysis, assuming that each chain bears one aldehyde group as chain end. As a general trend, calculated $\bar{\mathrm{DP}}$_n_ from NMR are in good correlation with SEC results. The formed polyazomethines have good thermal stability, with degradation temperatures above 350 °C, and even close to 400 °C (**P1**, Table 1). Lower degradation temperatures at 10% for **P6** could be explained by the presence of oligomers and remaining monomer within the sample (see Supporting Materials for TGA analyses and SEC traces).

### 3.2. Optical Characterization of Polyazomethines

The polyazomethines were characterized by absorbance spectroscopy in methylene chloride. Their spectra are compared in Figure 2.

As a general trend, polyazomethines with pendant methyl groups are slightly blue-shifted compared to their 2-ethylhexyl-bearing homologues. This result was expected, as DVM is blue shifted compared to DVEH (See Appendix A). The 2-ethylhexyl pendant groups might induce more planarity to the polyazomethine’s backbone or enabling better interdigitation, inducing a bathochromic shift [34]. This shift is relatively small, 4 nm at most between **P5** and **P6** and none in the case of **P1** and **P2**. For the latter polyazomethines, the presence of pendant *n*,*n*-dioctyl group on the fluorene moiety impedes the influence of shorter 2-ethylhexyl pendant group.

The intrinsic chemical structure of the fluorene moiety brings more planarity and a longer conjugation pathway to the polyazomethines, leading to a bathochromic shift and the highest absorbance maximum of the polyazomethines studied (See Table 2). The second-most red-shifted polyazomethines are the ones with a para-phenylene moiety (**P3** and **P4**). Indeed, the latter has only one aromatic ring and thus a shorter conjugation pathway. The most blue-shifted polyazomethines are the ones with meta-phenylene diamine moiety (**P5** and **P6**) which is explained by the meta bond, which interrupts conjugation and leads to a twist in the backbone.

The maximum absorbance is red-shifted in films in comparison to the absorbance in solution. This bathochromic shift is due to a rearrangement of the polyazomethine chains in films, as the latter can become more planar than in solution, inducing higher π-stacking and extended π-conjugation length. It appears that the alkyl side groups have an effect on this bathochromic shift: in film, **P2**, with a pendant methyl group, is more red-shifted than **P1** while **P5** is more red-shifted than **P6** (Table 2). The 2-ethylhexyl moiety on **P1**, combined with the alkyl groups of the fluorene moiety may create more constraint in the polymer backbone. For **P3** and **P4**, the pendant alkyl groups do not have an impact on the absorbance maximum in films. Unfortunately, all the polyazomethines studied exhibit a weak fluorescence in solution and no quantum yield could be measured (Appendix A). However, a particular behavior was observed for the fluorescence in films (Figure 3). Indeed, the most intense emission was obtained at a wavelength the polyazomethines barely absorb. This behavior suggests the formation of a charge transfer complex and could also be due to aggregates or excimers [35].

### 3.3. Model Molecules: Synthesis and XRD Analysis

To better understand the optical and physical properties of the vanillin-based polyazomethines, model compounds were designed (Figure 2). Vanillin and divanillin derivatives were reacted with a diamine or a monoamine respectively to prepare dimers or trimers as models of the polyazomethine backbone, allowing us to gain more insight on the impact of substituent position on opto-electronic properties.

These molecules were purified by recrystallization (see Experimental part for experimental conditions) and recovered with high purity enabling characterization by single-crystal X-Ray Diffraction (X-RD). The XRD analyses allowed us to unequivocally prove that azomethine models had a *E* (trans) configuration. For **M3a**, the angle between the plane of the phenyl ring and the one of the aniline moiety (noted θ_1_) is 69.1° (see Figure 4). Burgi et al. [36] solved the structure of benzylideneaniline and found a θ_1_ of 65.5. This twist could be due to the interactions between the azomethine bond and the ortho-substituents on the phenyl group, in the latter case the hydrogens, as was proposed by Bolduc et al. [37] Comparatively, the vinylene counterpart of benzylideneaniline is coplanar [38]. The θ_1_ angle is even lower for **M3b**, as there are two types of molecules in the lattice, linked together by interaction between the phenol of one molecule and the azomethine bond of the other one, giving more planarity to the molecules as attested by the values of θ_1_ equal to 39.7° and 15.7° respectively (see Supporting Materials for .cif file).

The same result was observed for **M1a** and **M1b**, as **M1a** has θ_1_ of 42.4° and **M1b** of 35.2°. However, in the case of **M1a,** the lattice is composed of two molecules linked on the side by interactions between the oxygen and the hydrogen of the vanillin rings. Therefore, the vanillin rings are nearly parallel or coplanar with each other. Moreover, with its relative planarity and symmetry, **M1a** can achieve π stacking, as two molecules can stack neatly on top of each other, which was not observed for **M1b** (See Figure 5).

**M2** has low θ_1_, 26.2° and 20.9° for the two azomethine bonds respectively. However even if they are more planar as they have a lower θ_1_, the **M2** molecules in the lattice twist around each other, as is shown in Figure 6. This could explain the poor solubility of the corresponding polymer, as the chains might twist around each other helicoidally.

The twist between the two aromatic rings of the divanillin is a torsion angle of 126.9°. **M4** has θ_1_ values of 57.1° and 54.7° for each azomethine bond, respectively (see Figure 6). In the lattice, the **M4** molecules are arranged two by two and twist around each other (see Figure 7). Moreover, some interactions between the nitrogen of the azomethine function and the hydrogens on the divanillin rings may occur. These interactions suggest the formation of a charge transfer complex between the two **M4** molecules, generating a dipole with the nitrogen of the azomethine acting as an acceptor and the divanillin ring acting as a donor.

### 3.4. Optical Characterization of Model Molecules

The absorbance spectra of the model compounds and corresponding polyazomethine are represented in Figure 8.

As expected, **M1a** is more red-shifted than **M3a** as **M1a** has a longer conjugation pathway with its three aromatic rings. However, **M3a** and **M4** have the same maximum of absorbance. One could expect **M4** to be slightly shifted, as it corresponds to twice **M3a,** but this was not observed. There is actually a slight change on the second peak of the absorbance spectrum, at 280 nm; this peak is comparatively more intense for **M4** than for **M3a**. **M1a** has also the same absorbance maximum as **P4**. Again, one could expect **P4** to be more red-shifted, as it corresponds to a polymer chain with a longer conjugation pathway, but this is not the case. Indeed, the absorbance spectra of **M1a** and **P4** are very similar. It can be explained by a limitation of conjugation, due to the link, in meta position, between the aromatic rings of the divanillin and the azomethine bond, in the polymer and in the model compound as well. This is the reason why **M3a** and **M4** have the same absorbance maximum even if **M4** is twice longer than **M3a**. the two halves being linked by a meta bond, which dramatically lowers and even breaks the conjugation. This is also true for **P4** for which the length of the conjugation pathway is dictated by the bond between the aromatic rings of the divanillin, which is in meta with respect to the azomethine bonds, and therefore **P4** has the same absorbance as **M1a**.

## 4. Conclusions

In conclusion, we report the synthesis and characterization of vanillin-based polyazomethines and model compounds. The polyazomethines were synthesized under micro-wave irradiation for only 5 min, followed by a recovery step using a rotary evaporator. The key step during the synthesis is actually the recovery step, during which molar masses are greatly improved. This experimental protocol was successfully applied to various diamines and divanillin derivatives, leading to polyazomethines with molar masses up to 21 kg/mol.

As for the optical properties, the absorbance maximum of the polyazomethines can be tuned by changing the diamine as well as the nature of the pendant alkyl group of the divanillin. However, the polyazomethines are not fluorescent in solution, but exhibit an original behavior in solid state (as films) feature explained by the formation a charge transfer complex.

The structure/properties study of vanillin-based azomethines used as models allowed us to unequivocally confirm the E configuration though XRD and to highlight the cross-conjugated nature of divanillin-based polyazomethine. Further studies are needed to confirm the presence of charge-transfer behavior in solid state. Furthermore, the doping of these materials will be also studied in order to improve fluorescence.

## Data Availability

All Crystal Data were deposited on Cambridge Crystallographic Data Center (CCDC), Cambridge, UK. Deposition numbers are listed in the Appendix A.

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
