# Peer review of "Synthesis and Characterization of Vanillin-Based π-Conjugated Polyazomethines and Their Oligomer Model Compounds"

_molecules, 2022, doi:10.3390/molecules27134138_

Round 1
Reviewer 1 Report
Comments to Authors
Title: Synthesis and Characterization of Vanillin-based π-Conju-2 gated Polyazomethines and Their Oligomer Model Compounds
Authors: Lauriane Giraud, Stéphane Grelier, Etienne Grau, Laurent Garel, Georges Hadziioannou, Brice Kauffmann, Éric Cloutet, Henri Cramail and Cyril Brochon
Journal: Molecules
Manuscript ID: Molecules-1715416
In the presented study, after synthesizing the divanillin compound, poly azomethines were obtained from condensation reactions with aromatic diamine compounds. The structure confirmation of all the compounds obtained was done by different spectral methods. However, the signal of the proton belonging to the aromatic -OH group of the DV compound does not appear in the spectrum in Fig.S1. The 1H NMR measurement of DV should be taken again. The Mw values of the polystyrene standards used for the calibration curve should be given. In addition, some measurements should be taken about the applications of these polymers.
As result, in my opinion, this manuscript can accept for publication in Molecules from major revision.

Author Response
Dear reviewer 1. Thank you for your remarks and comments. We tried to take into account all of them in the revised document. Below you will find our answers in italic.
- General appreciations: Improvement of cited references
Introduction was improved, and 2 recent references have been added (ref 9 and 14)
- General appreciations: Clarity of discussion
Discussion was clarified and improved pages 14; 15 and 26
Comments:
1/ “the signal of the proton belonging to the aromatic -OH group of the DV compound does not appear in the spectrum in Fig.S1. The 1H NMR measurement of DV should be taken again”.
NMR spectra were recently recorded, and H from hydroxyl is still difficult to see. It is commonly observed in chloroform, and actually it was already the case in previous works (Garbay et al ans Llevot et al : ref 8 and 10 to 12).
2/ “The Mw values of the polystyrene standards used for the calibration curve should be given.”
We add the 11 standards molecular weigt values, Page 5.
3/ “In addition, some measurements should be taken about the applications of these polymers.”
Quantum yield and fluorescence measurement have been made. QY are low and we decided to not include them in this paper which is more about synthesis. Further physical characterization (e.g. Charge mobility) will be included in a further paper.
Reviewer 2 Report
Thank you for inviting me to review this article. In the present study, the authors are focused on “Synthesis and Characterization of Vanillin-based π-Conju-2 gated Polyazomethines and Their Oligomer Model Compounds”. On the whole, it is interesting and certainly worth investigating. But to meet the increasingly high-quality standard of Molecules, a careful revision is absolutely needed before the acceptance.
- pi-conjugated should be π-conjugated.
- Why they didn’t try other solvents and catalysts?
- Please neatly edit the manuscript, and pay attention to the use of punctuation marks and spaces.
- There are several grammatical mistakes, please correct them.
In conclusion, the results reported in this work are interesting and suitable for publication in the Molecules after minor/major correction.
Author Response
Dear reviewer 2. Thank you for your remarks and comments. We tried to take into account all of them in the revised document. Below you will find our answers in italic.
- pi-conjugated should be π-conjugated.
It has been corrected
- Why they didn’t try other solvents and catalysts?
to the exp conditions of polymerization (130°C) we didn’t use low boiling point solvent. Since we didn’t have solubility issue, we didn’t try more exotic solvent such as DMSO or Chlorobenzene. However, dimers and trimers are obtained in reflux methanol. We try it (once) for polymerization directly in the rotavapor : we obtain polymers with moderate Mw. For the catalyst, we needed on organic acid, we choose the classical PTSA, and we never others. Nevertheless, in the preliminary study, we try without, just with silica (which is acidic) and it works but with longer polymerization times.
- Please neatly edit the manuscript, and pay attention to the use of punctuation marks and spaces.
We made several correction, we did our best
- There are several grammatical mistakes, please correct them.
Once again, we made several corrections.
Round 2
Reviewer 1 Report
Comments to Authors
Title: Synthesis and Characterization of Vanillin-based π-Conju-2 gated Polyazomethines and Their Oligomer Model Compounds
Authors: Lauriane Giraud, Stéphane Grelier, Etienne Grau, Laurent Garel, Georges Hadziioannou, Brice Kauffmann, Éric Cloutet, Henri Cramail and Cyril Brochon
Journal: Molecules
Manuscript ID: Molecules-1715416R1
In the presented study, after synthesizing the divanillin compound, poly azomethines were obtained from condensation reactions with aromatic diamine compounds. The structure confirmation of all the compounds obtained was done by different spectral methods.
Still, the signal of the proton belonging to the aromatic -OH group of the DV compound does not appear in the spectrum in Fig.S1. The 1H NMR measurement of DV should be taken again. The PD1 values given for P4 and P6 in Table 1 are too high. Although the Mn values are very low, these values being observed so high indicates that there is a measurement problem. SEC measurements of P4 and P6 should be repeated.
As result, in my opinion, this manuscript can accept for publication in Molecules from major revision.
author responses:
Dear reviewer. Thank you for your remarks and comments. We tried to take into account all of them in the revised document. Below you will find our answers in italic.
In the presented study, after synthesizing the divanillin compound, polyazomethines were obtained from condensation reactions with aromatic diamine compounds. The structure confirmation of all the compounds obtained was done by different spectral methods.
Still, the signal of the proton belonging to the aromatic -OH group of the DV compound does not appear in the spectrum in Fig.S1. The 1H NMR measurement of DV should be taken again.
As we said previously, phenolic protons are difficult to see (because of their lability), however NMR has been done again in D6 DMSO. We saw a small peak at 10.25 ppm which can be attributed to this aromatic proton, and also water at 3.5 ppm. Fig. S1 has been updated, with a larger scale.
We also performed NMR in D6 DMSO after addition of a small amount of D2O, and actually it disappeared.
Below, we draw the two spectra: before (black) and after D2O addition(grey), showing the disappearance of the peak at 10.25 ppm
The PD1 values given for P4 and P6 in Table 1 are too high. Although the Mn values are very low, these values being observed so high indicates that there is a measurement problem. SEC measurements of P4 and P6 should be repeated.
We did again the measurements with a sharper integration (table in the manuscript has been updated for P4 and P6) but it doesn’t change so much. In this 2 cases, there is a lot of oligomers, chromatograms are very broad as you can see in figures S14 and S17. This particularly true for P6, in that case integration is very difficult.